# Analysis of change in patient-reported outcome measures with floor and ceiling effects using the multilevel Tobit model: a simulation study and an example from a National Joint Register using body mass index and the Oxford Hip Score

Adrian Sayers ![ORCID],[1,2] Michael R Whitehouse ![ORCID],[1,3] Andrew Judge,[1,3] Alex J MacGregor,[4] Ashley W Blom,[3] Yoav Ben-Shlomo[2]

For numbered affiliations see end of article.

**Correspondence to**
Adrian Sayers;
adrian.sayers@bristol.ac.uk

## ABSTRACT

**Objectives** This study has three objectives. (1) Investigate the association between body mass index (BMI) and the efficacy of primary hip replacement using a patient-reported outcome measure (PROMs) with a measurement floor and ceiling, (2) Explore the performance of different estimation methods to estimate change in PROMs score following surgery using a simulation study and real word data where data has measurement floors and ceilings and (3) Lastly, develop guidance for practising researchers on the analysis of PROMs in the presence of floor and ceiling effects.

**Design** Simulation study and prospective national medical device register.

**Setting** National Register of Joint Replacement and Medical Devices.

**Methods** Using a Monte Carlo simulation study and data from a national joint replacement register (162 513 patients with pre- and post-surgery PROMs), we investigate simple approaches for the analysis of outcomes with floor and ceiling effects that are measured at two occasions: linear and Tobit regression (baseline adjusted analysis of covariance, change-score analysis, post-score analysis) in addition to linear and multilevel Tobit models.

**Primary outcome** The primary outcome of interest is change in PROMs from pre-surgery to 6 months post-surgery.

**Results** Analysis of data with floor and ceiling effects with models that fail to account for these features induce substantial bias. Single-level Tobit models only correct for floor or ceiling effects when the exposure of interest is not associated with the baseline score. In observational data scenarios, only multilevel Tobit models are capable of providing unbiased inferences.

**Conclusions** Inferences from pre- post-studies that fail to account for floor and ceiling effects may induce spurious associations with substantial risk of bias. Multilevel Tobit models indicate the efficacy of total hip replacement

## Strengths and limitations of this study

► We use a comprehensive simulation study and large prospective study set to investigate the effect of floor and ceiling effects in the analysis of change in patient-reported outcome measure pre- post-surgery.
► We demonstrate the use and performance of mutlilevel Tobit models to estimate change in patient-reported outcome measures with floor and ceiling effects and compare them to simple analytical approaches.
► We compare and demonstrate a variety of estimators in simulation under a variety of different data generating mechanisms and compare results to real world data.
► This is the largest and most comprehensive analysis of the effect of body mass index on the efficacy of total hip replacement and provides data which will influence the provision of hip replacement.

is independent of BMI. Restricting access to total hip replacement based on a patients BMI can not be supported by the data.

## INTRODUCTION

In many non-randomised experiments, researchers are interested in assessing how change in health status is associated with a covariate of interest. While there is much guidance available on assessing change in randomised experiments, and extensive discussion with respect to efficiency and bias,[1–9] the guidance in non-randomised studies is less clear. The principle difference is that in observational studies we do not expect balance

between different levels of an exposure at baseline, in addition to expecting imbalance in other confounding factors. Glymour *et al* advocate the use of, Simple Analysis of Change Scores (SACS) without baseline adjustment to achieve unbiased causal effect estimates using causal arguments presented through Directed Acyclic Graphs.[10] They briefly suggest that in settings with floor and/or ceiling effects, that standard change analyses with and without baseline adjustment are both biased, and non-standard analyses based on Tobit models (censored regression) may ameliorate floor and ceiling problems. The degree to which Tobit models ameliorate the problems caused by floor and ceiling effects is unclear. Some authors suggest that using percentage change is one strategy to avoid dealing with floor and ceiling effects, but Twisk highlighted that this simply represents a linear transformation of change,[11] and therefore, does not deal with the problem of floor and ceiling effects. Twisk and Rijmen also describe the use of a longitudinal (multilevel) Tobit regression model to appropriately account for floor and ceiling effects in studies with repeated measures.[12] However, since its publication in 2009, there have only be a handful of analyses that use multilevel Tobit models (MLTMs),[13–16] suggesting that lack of familiarity with these methods or understanding of when they can and should be applied has deterred analysts in their use, or when they can be applied.

MLTMs are now incorporated in mainstream statistical software packages, such as Stata version 15. Given their accessibility, they could arguably be used more frequently than they are. This is relevant considering that the use of measurement instruments with floor and ceiling effects are omnipresent in health-related research. Examples include outcomes in health-related quality of life (eg, EQ-5D, SF-36 and SF-12), psychological well-being (eg, Hospital Anxiety and Depression Scale, Edinburgh Postnatal Depression Scale) and disease-specific measures of well-being (eg, Western Ontario and McMaster Universities Osteoarthritis Index and Oxford Hip Score (OHS) as used in patients with osteoarthritis (OA)). Despite this, there is very little guidance available with respect to the consequences of using measurement instruments with floor or ceiling effects, when attempting to make inferences about the effect of an exposure on the change (between two time points) of an outcome of interest.

In this paper, we use a Monte Carlo (MC) simulation study to compare the performance of multilevel linear and Tobit models, ordinarily least squares (OLS) regression and single-level Tobit regression, with and without adjustment for baseline scores, in the analysis of change in three different non-randomised experiments and a randomised experiment. We also demonstrate the use of these models using real world data from a large national joint replacement register.

We motivate the simulation and exemplar data analysis using an example from joint replacement research describing the association between body mass index (BMI) and the change in a disease specific patient-reported outcome measure (PROM), the OHS. The issue is contentious in the UK[17–19] and USA[20] as some organisations suggest restricting joint replacement to patients based on their BMI, citing an increased risk of revision surgery and lack of efficacy of surgery. The small increase in absolute risk of revision in obese patients, must be balanced against the other benefits of joint replacement, including a reduction in pain and improved physical functioning. Therefore, it is of interest to clinicians, policy-makers and patients to know the relative effect of obesity on the efficacy of total hip replacement (THR) compared with 'normal weight patients'.

## METHODS

### Simulation study aims

We investigated the performance of four different methods of analysis, when estimating the effect of an exposure (BMI) on change in response (PROM) before and after THR with floor and ceiling effects using the Aims, Data Generating Process (DGP), Methods, Estimand, Performance approach recommended by Morris *et al.*[21]

### Data generating process

We simulated longitudinal data of 'well-being' before and after surgery. We assume that 'well-being' is a latent, truly continuous and stable construct which is measured imperfectly by the OHS. Measurement error and floor/ceiling effects are then added to the latent construct to illustrate their consequences.

We assume the response, well-being, is a latent construct $(y_{ij}^*)$ measured at the $i^{th}$ occasion, where $i$ varies from 0 (pre-surgery) to 1 (1-year post-surgery), for the $j^{th}$ individual is modelled as a linear function of time. $x_{0j}$ is mean-centred BMI categories according to WHO criteria, that is, $-2=$ BMI $<18.5$ kg/m$^2$ (under weight), $-1=$ $18.5<$BMI$\leq25$ kg/m$^2$ (normal), $0 = 25<$BMI$\leq30$ kg/m$^2$ (overweight), $1 = 30<$BMI$\leq35$ kg/m$^2$ (obese), and $2=$ BMI $>35$ kg/m$^2$ (morbidly obese), i.e. $x_{0j} = 0$ is a patient with a BMI classed as overweight.

$$y_{ij}^* = \beta_0 + u_{0j} + \left(\beta_1 + u_{1j}\right) t_{ij} + \beta_2 x_{0j} + \beta_3 x_{0j} t_{ij}$$

$$\begin{bmatrix} u_{0j} \\ u_{1j} \end{bmatrix} \sim N\left(0, \Omega_u\right), \Omega_u = \begin{bmatrix} \sigma_{u0}^2 & \\ \sigma_{u01} & \sigma_{u1}^2 \end{bmatrix}$$

where $t_{ij}$ is the time at which measurement $i$ was taken on individual $j$, coded as 0 at pre- and 1 post-surgery. $\beta_0$ is the baseline population average response for a patient with average BMI, and $u_{0j}$ represents the $j^{th}$ individual difference from the baseline response. The sum of $\beta_0 + u_{0j}$ is the individual baseline response for a patient with average BMI. $\beta_1$ represents the population average change per unit increase in time for a patient with average BMI, and $u_{1j}$ represents the $j^{th}$ individual difference from the population average change per unit increase in time. The sum $\beta_1 + u_{1j}$ is the individual average change per unit increase in time for a patient with average BMI. $\beta_2$ represents the effect of a 1-unit increase in the exposure $(x_{0j})$ of interest (BMI) pre-surgery and $\beta_3$ represents the effect of a 1-unit

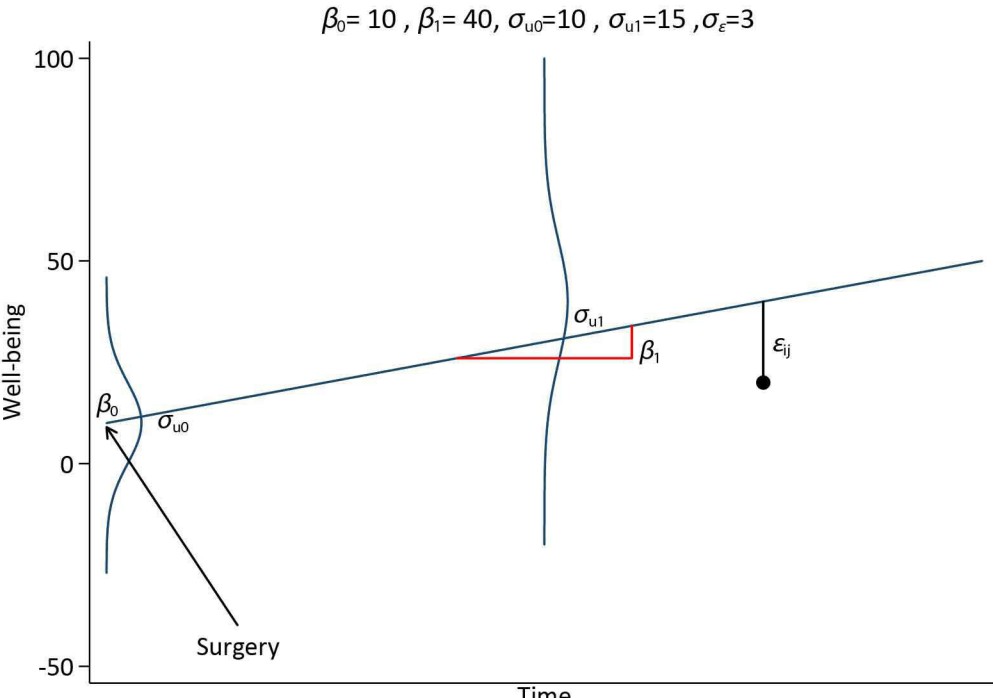

**Figure 1** Graphical illustration of a multilevel random intercept and slope model used to generate data for a individual with average BMI. BMI, body mass index.

increase in BMI ($x_{0j}$) on the pre- post-surgery change in well-being ($y_{ij}^*$). The variance in individual deviations from the population average response at baseline and the average rate of change are $\sigma_{u0}^2$ and $\sigma_{u1}^2$, respectively. The covariance between baseline measurements and rate of change is characterised by $\sigma_{u01}$ (with correlation $\rho_{u01}$).

Under the assumption of linear change, data were simulated from a multilevel model (MLM) with a random intercept and slope, see figure 1 for an illustration of a patient trajectory with an average BMI.

The observed response without floor and ceiling effects ($y_{ij}$) is simulated by adding measurement error in the linear trajectory, $\varepsilon_{ij}$, to the latent response, where $\varepsilon_{ij} \sim N(0, \sigma_\varepsilon^2)$.

$$y_{ij} = y_{ij}^* + \varepsilon_{ij}$$

A response with floor and ceiling effects $y_{ij}^{FC}$ is simulated by restricting the response to lie between 0 and 48.

$$y_{ij}^{FC} = \begin{cases} 0 & \text{if } y_{ij} \leq 0 \\ y_{ij} & \text{if } 0 < y_{ij} < 48 \\ 48 & \text{if } y_{ij} \geq 48 \end{cases}$$

See figure 2 for a graphical illustration of the trajectory generation: we first simulate $y_{ij}^*$, then add some measurement error ($\varepsilon_{ij}$) to yield an observed response ($y_{ij}$), and finally add floor and ceiling effects to obtain the observed truncated response ($y_{ij}^{FC}$).

We compared four DGPs to illustrate a range of scenarios by manipulating $\beta_2$, $\beta_3$ and $\rho_{u01}$ ($\sigma_{u01}$) to influence the association between pre-surgery and post-surgery outcomes. $\beta_0$, $\beta_1$, $\sigma_{u0}$, $\sigma_{u1}$, and $\sigma_{\varepsilon ij}$ were fixed at 10, 40, 10, 15 and 3, respectively. DGP 1 is a null model,

where there is a baseline effect of the exposure is $\beta_2=-3$, but the exposure did not influence change over time ($\beta_3=0$), and there is no correlation between baseline values and subsequent change ($\rho_{u01}=0$).

DGP 2 replicates a simple randomised trial where there is no difference between levels of the exposure at baseline ($\beta_2=0$), but the exposure did influence change over time ($\beta_3=-3$), and there is no correlation between baseline values and subsequent change ($\rho_{u01}=0$). DGP 3 and DGP 4 replicate a cohort study, where there is a difference between levels of the exposure at baseline ($\beta_2=-3$), and the exposure also influenced change over time ($\beta_3=-3$). DGP 3 specified no correlation between baseline values and subsequent change ($\rho_{u01}=0$), whereas DGP 4 specified a negative correlation between baseline values and change ($\rho_{u01}=-0.5$), reflecting the fact the joint replacement surgery has the tendency to normalise an individuals well-being, see figure 3 for an illustration of the associated trajectories.

We conducted an MC simulation with 1000 replicated datasets, each with 10 000 patients. A balanced dataset, that is, three data points for each individual, was simulated to ensure identification of the linear and Tobit MLMs occurred, that is, two data points allow estimation of baseline and change parameters but not measurement error. The middle data point was then dropped to replicate a pre- post- study design.

### Method of analysis

For data sets with three measurement occasions, a linear MLM and an MLTM that reflects the DGP were fitted to the data, see equation 1.

In datasets with two measurement occasions, that is, a pre- post- study design, single-level OLS and Tobit models

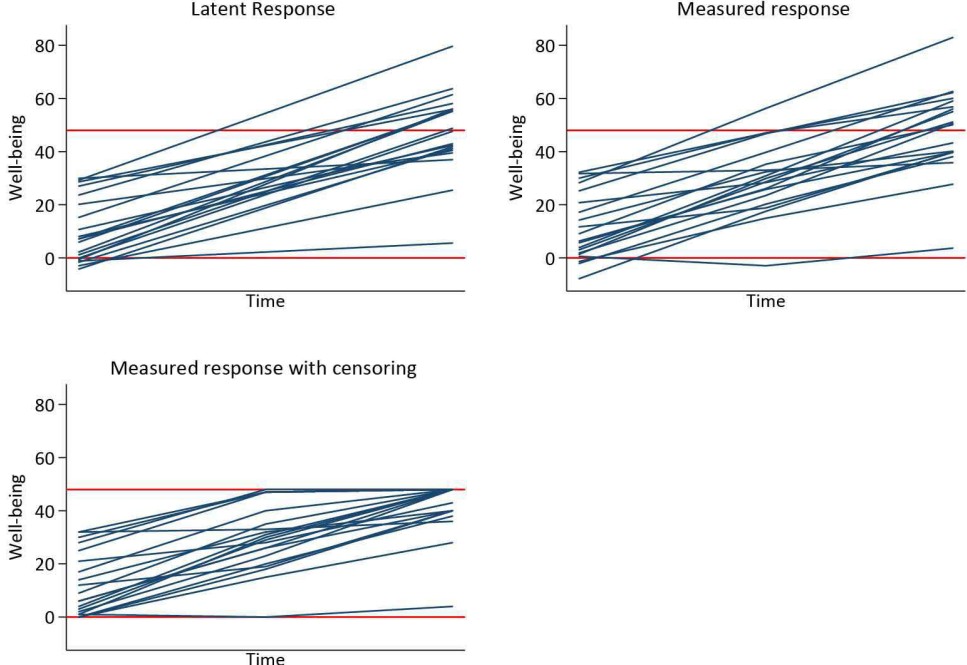

**Figure 2** Graphical illustration of the data generating process of the latent, measured and measured response with floor and ceiling effects. The latent response is $y_{ij}^{*}$, the measured response is $y_{ij}$, and the measured response with censoring is $y_{ij}^{FC}$.

were fitted to the data. Tobit models were only used when floor and ceiling effects had been simulated. Three different models were explored:

1. A simple model for post-surgery well-being.

$$y_{1j}^{FC} = \alpha_1 + \alpha_2 x_{0j} + \varepsilon_j$$

2. A SACS.

$$\left( y_{1j}^{FC} - y_{0j}^{FC} \right) = \alpha_6 + \alpha_7 x_{0j} + \varepsilon_j$$

3. A model for change adjusted for baseline, that is, baseline adjusted analysis of covariance (ANCOVA). This model is equivalent to a model for the post-score adjusted for baseline ANCOVA, with the exception of the interpretation of the intercept.

$$\left( y_{1j}^{FC} - y_{0j}^{FC} \right) = \alpha_8 + \alpha_9 x_{0j} + \alpha_{10} y_{0j}^{FC} + \varepsilon_j$$

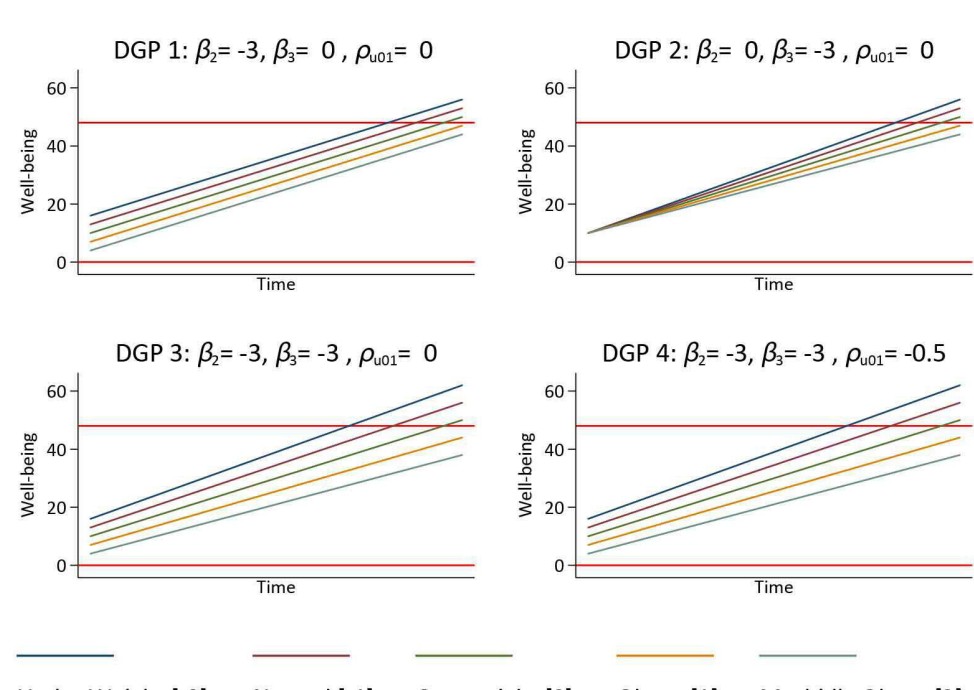

**Figure 3** Graphical illustration of the four DGP used to investigate the effect of floor and ceiling effects on analysis of pre-surgery and post-surgery change with BMI as an exposure. Horizontal red lines at 0 and 48 indicate floor and ceilings of the measurement instrument. BMI, body mass index; DGP, data generating process.

In addition, an underidentified MLTM, equivalent to equation 1 with constrained error variance $\sigma_\varepsilon^2$ was fitted in the spirit of a sensitivity analysis, where $\sigma_\varepsilon^2$ was constrained to a value from 5, 10, 15, 20, 25 and 30.

### Estimand

The estimand of interest is the population average effect of the interaction between the exposure and change in slope, that is, $\beta_3$ the pre- post-surgery change in well-being. We test whether the exposure modifies the improvement post-surgery (ie, the null hypothesis that $\beta_3 = 0$).

### Performance

The performance of each method was explored in terms of bias, coverage, empirical SE, model-based SE, mean square error, relative error and relative precision.

### National Joint Registry of England, Wales, Northern Ireland and the Isle of Man

Using data from the National Joint Registry (NJR), we investigated the association between BMI and a PROM, the OHS, in patients undergoing elective THR between 1 April 2003 and 22 February 2017.

### Data source

The NJR commenced data collection in April 2003; at inception it was mandatory for all THRs conducted in the private sector to be entered into the NJR, and from 2011 all THR procedures in the public and private sector were required to be entered into the NJR. A recent national audit of data entered into the NJR between 2014 and 2015 estimated data capture of 95% for primary THR and 91% for revision THR.

### Inclusion/exclusion criteria

All consenting patients undergoing THR were eligible to be included in the analysis. Patients were included if their patient history was unique and consistent, that is, contained no duplicates, revision prior to primary, or currently held in query by the submitting unit. Due to the requirement for reliable date information, patients who were indicated to have died prior to undergoing a procedure, were more than 110 years of age, had undergone a procedure prior to their date of birth, or received a procedure prior to 2003 were excluded from the analysis. Only primary THRs, where the primary indication for operation was OA with unique prosthesis combinations were included in the analysis. All THRs with metal-on-metal bearing combinations were excluded from the analysis due to the exceptionally high failure rate in this group.[22 23] Patients who were less than 50 years of age at the date of the index THR were also excluded, due to the high likelihood that these cases are due to OA secondary to other pathology.

See online supplementary figure 1 for a detailed breakdown of inclusion criteria.

### Primary exposure

The primary exposure of interest in this study is BMI. BMI was introduced into the second 'Minimal Data Set' in 2004. Patients with BMI between 10 and 60 were included in the analysis. BMI measures were excluded as implausible if height and weight measures were less than 130 cm and weight less than 30 kg, respectively. See online supplementary figure 2.

### Primary outcome

The primary outcome of interest in this study is change in OHS after surgery. Linked National PROMs were first available in 2009, see online supplementary figure 3 for details of linkage.

### Confounding factors

Preoperative confounding factors were thematically organised into groups: (1) Patient factors included sex, American Society of Anesthesiologists grade and operation funder; (2) Operation factors included fixation, approach, patient position during surgery, anaesthetic type, thromboprophylaxis regimen, bearing and year of primary THR; (3) The setting of the treatment episode (ie, private or National Health Service hospital); (4) Consultant-based factors included the training status of the primary surgeon performing the operation and (5) Deprivation factors were based on the English indices of multiple deprivation (an area-based index of deprivation).

### Statistical analyses

Means, SD and IQR points were used to describe continuous variables. Frequencies and percentages were used to describe categorical variables.

The association between change in PROMS score was investigated using the same single-level methods and the ML Tobit model with constrained error variances described in the simulation study as an exemplar. In addition, we conducted more comprehensive analyses using restricted cubic splines (RCS) to model the BMI association in the ML Tobit model with constrained error variance, single-level linear and Tobit SACS, ANCOVA and post-score models. In the ML Tobit model, BMI was modelled with RCS at baseline and its interaction with time. Correspondingly, we adjusted OHS for patient and deprivation confounding factors at baseline and operation, setting and confounding factors with an interaction with time, that is, operative factors and settings influence the change in outcome but not the baseline response. In single-level models, the effect of BMI was modelled using RCS and adjusted for confounding factors using standard regression approaches.

### Missing data

Due to the method of data collection in the national PROMS programme, item non-response is masked. De facto mean imputation of up to two missing items in the OHS occurred automatically. In addition, despite valid values appearing with individual OHS items, if the questionnaire was marked as 'not complete', implausible

overall scores were obtained. For simplicity, only patients with complete pre-operative and post-operative PROMS were used in the analysis. BMI is missing in a substantial proportion of the cohort. Patients prior to 2004 did not have BMI recorded, and the proportion of patients with missing BMI in 2004 is large. In 2009, ~40% of patients did not have BMI recorded; this reduced year on year and in 2016 was ~18% of eligible patients.

For pedagogical simplicity, we use complete-case analyses throughout.

### Patient and public involvement

Patient representatives sit on the committee structure of the NJR. The research priorities of the NJR are identified by this committee structure and approved by the patient representatives. Patients were not involved in the setting of the research question or the outcome measures, nor were they involved in designing or implementing this work or interpretation of the results. We are unable to disseminate results of this study directly to study participants due to the anonymous nature of the data. We plan to disseminate our findings to the NJR, via their communications team, to relevant individuals with regard to the provision of joint replacement and to the general population through the local and national press.

### RESULTS
#### Simulation study

Figure 4 illustrates the results from the MC simulation for each DGP. It is clear that MLM, OLS methods and in DGP's 1, 3 and 4 (observational scenarios) single-level Tobit models all exhibit substantial bias. Only the ML Tobit with three data points provides unbiased estimates in all scenarios. Constrained ML Tobit models are close to being unbiased, but slightly over estimate the effect size, see table 1, Single-level Tobit models also provide unbiased estimates for DGP 2 (the randomised trial). Empirical SE, mean squared error, relative error and relative precision for each of the methods are reported in online supplementary table 1.

Figure 5 illustrates the spread of model-based (SEs for each method by DGP. It is clear that the variation and absolute magnitude of SE in MLM with three data points per person is less than that of ML Tobit models. Similarly, model-based SEs from OLS methods are smaller and less variable than single-level Tobit methods. In DGP 2, the randomised trial, it is interesting to note that the SE from Tobit ANCOVA models are marginally smaller than for Tobit SACS. While there is little difference in terms of bias from the constrained ML Tobit models, see figure 4, the size and variability of estimated SEs increased with increasing value of the constrained of $\sigma_\varepsilon^2$.

Online supplementary figures 4–15, illustrate the coverage of 95% CIs in each DGP. Unsurprisingly, coverage of methods which demonstrate bias is very poor, while coverage is at nominal levels for the ML Tobit model with three data points. The results from constrained ML Tobit indicate coverage less than the nominal levels. Coverage less than the advertised levels is principally due to the bias in estimate. However, when the estimates from the model are unbiased, as in DGP 2 with $\sigma_\varepsilon^2 = 5$, coverage

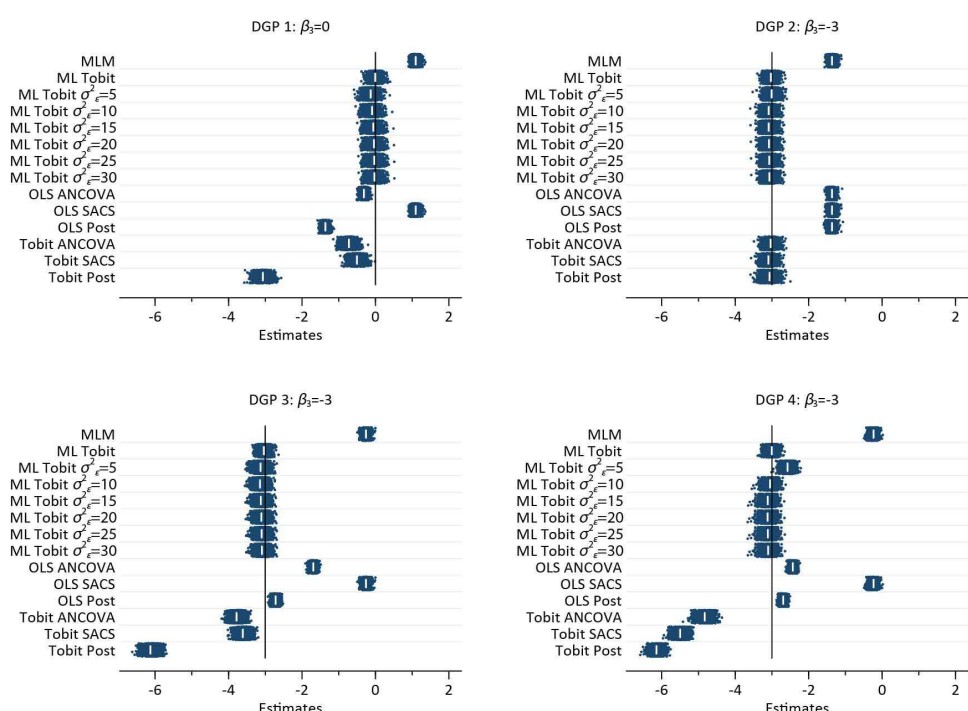

**Figure 4** Plot of 1000 estimates by each DGP, for each method of analysis. Within each method, the vertical axis is the repition number of each simulated dataset. The white pipe symbol is the average of the estimates. ANCOVA, analysis of covariance; DGP, data generating process; MLM, multilevel model; OLS, ordinarily least squares; SACS, Simple Analysis of Change Scores.

**Table 1** Simulation estimates of performance characteristics including mean and Monte Carlo Standard Error in parantheses of different models using each DGP.

| Model | DGP 1: $\beta_3 = 0$ | | DGP 2: $\beta_3 = -3$ | | DGP 3: $\beta_3 = -3$ | | DGP 4: $\beta_3 = -3$ | |
|---|---|---|---|---|---|---|---|---|
| **Estimate** | | | | | | | | |
| MLM | 1.1 | (0.0024) | −1.36 | (0.0023) | −0.26 | (0.0024) | −0.23 | (0.0025) |
| ML Tobit | −0.0056 | (0.0038) | −3.03 | (0.0037) | −3.04 | (0.0037) | −3.01 | (0.0037) |
| ML Tobit $\sigma_\epsilon^2 = 5$ | −0.13 | (0.0044) | −3.01 | (0.0042) | −3.13 | (0.0046) | −2.57 | (0.0038) |
| ML Tobit $\sigma_\epsilon^2 = 10$ | −0.093 | (0.0044) | −3.09 | (0.0042) | −3.14 | (0.0045) | −3.05 | (0.0041) |
| ML Tobit $\sigma_\epsilon^2 = 15$ | −0.057 | (0.0044) | −3.09 | (0.0042) | −3.12 | (0.0045) | −3.11 | (0.0043) |
| ML Tobit $\sigma_\epsilon^2 = 20$ | −0.04 | (0.0044) | −3.08 | (0.0042) | −3.11 | (0.0045) | −3.12 | (0.0044) |
| ML Tobit $\sigma_\epsilon^2 = 25$ | −0.031 | (0.0044) | −3.07 | (0.0042) | −3.1 | (0.0045) | −3.11 | (0.0044) |
| ML Tobit $\sigma_\epsilon^2 = 30$ | −0.026 | (0.0044) | −3.07 | (0.0042) | −3.09 | (0.0045) | −3.1 | (0.0045) |
| OLS SACS | 1.1 | (0.0024) | −1.36 | (0.0023) | −0.26 | (0.0024) | −0.23 | (0.0025) |
| OLS ANCOVA | −0.31 | (0.0022) | −1.36 | (0.002) | −1.69 | (0.0021) | −2.43 | (0.0019) |
| OLS Post | −1.36 | (0.0023) | −1.36 | (0.0022) | −2.72 | (0.0023) | −2.69 | (0.0018) |
| Tobit SACS | −0.72 | (0.0044) | −3.04 | (0.0041) | −3.78 | (0.0044) | −4.83 | (0.0048) |
| Tobit ANCOVA | −0.5 | (0.0046) | −3.09 | (0.0042) | −3.61 | (0.0045) | −5.5 | (0.0042) |
| Tobit Post | −3.07 | (0.0049) | −3.06 | (0.0047) | −6.13 | (0.005) | −6.14 | (0.004) |
| **Coverage** | | | | | | | | |
| MLM | 0 | (0) | 0 | (0) | 0 | (0) | 0 | (0) |
| ML Tobit | 94.8 | (0.7) | 95.3 | (0.67) | 93.9 | (0.76) | 95.4 | (0.66) |
| ML Tobit $\sigma_\epsilon^2 = 5$ | 67.5 | (1.48) | 86.7 | (1.07) | 71.6 | (1.43) | 4.6 | (0.66) |
| ML Tobit $\sigma_\epsilon^2 = 10$ | 83.5 | (1.17) | 85 | (1.13) | 77.4 | (1.32) | 92.2 | (0.85) |
| ML Tobit $\sigma_\epsilon^2 = 15$ | 91.1 | (0.9) | 87.7 | (1.04) | 85.7 | (1.11) | 86.8 | (1.07) |
| ML Tobit $\sigma_\epsilon^2 = 20$ | 92.7 | (0.82) | 90 | (0.95) | 88.2 | (1.02) | 87.1 | (1.06) |
| ML Tobit $\sigma_\epsilon^2 = 25$ | 93.4 | (0.79) | 91.6 | (0.88) | 89.5 | (0.97) | 88.2 | (1.02) |
| ML Tobit $\sigma_\epsilon^2 = 30$ | 93.6 | (0.77) | 91.9 | (0.86) | 91.1 | (0.9) | 88.9 | (0.99) |
| OLS SACS | 0 | (0) | 0 | (0) | 0 | (0) | 0 | (0) |
| OLS ANCOVA | 0.8 | (0.28) | 0 | (0) | 0 | (0) | 0 | (0) |
| OLS post | 0 | (0) | 0 | (0) | 2.4 | (0.48) | 0 | (0) |
| Tobit SACS | 0.1 | (0.1) | 94.4 | (0.73) | 0 | (0) | 0 | (0) |
| Tobit ANCOVA | 7.2 | (0.82) | 89.7 | (0.96) | 1.2 | (0.34) | 0 | (0) |
| Tobit post | 0 | (0) | 93.3 | (0.79) | 0 | (0) | 0 | (0) |
| **Model SE** | | | | | | | | |
| MLM | 0.074 | (2E-05) | 0.074 | (2E-05) | 0.077 | (2E-05) | 0.078 | (2E-05) |
| ML Tobit | 0.12 | (3E-05) | 0.12 | (3E-05) | 0.12 | (3E-05) | 0.12 | (3E-05) |
| ML Tobit $\sigma_\epsilon^2 = 5$ | 0.1 | (3E-05) | 0.1 | (3E-05) | 0.11 | (3E-05) | 0.11 | (4E-05) |
| ML Tobit $\sigma_\epsilon^2 = 10$ | 0.12 | (3E-05) | 0.12 | (3E-05) | 0.13 | (4E-05) | 0.12 | (3E-05) |
| ML Tobit $\sigma_\epsilon^2 = 15$ | 0.13 | (4E-05) | 0.13 | (4E-05) | 0.14 | (4E-05) | 0.13 | (3E-05) |
| ML Tobit $\sigma_\epsilon^2 = 20$ | 0.13 | (5E-05) | 0.13 | (5E-05) | 0.14 | (5E-05) | 0.14 | (4E-05) |
| ML Tobit $\sigma_\epsilon^2 = 25$ | 0.14 | (5E-05) | 0.13 | (5E-05) | 0.14 | (6E-05) | 0.14 | (5E-05) |
| ML Tobit $\sigma_\epsilon^2 = 30$ | 0.14 | (5E-05) | 0.13 | (5E-05) | 0.15 | (6E-05) | 0.14 | (5E-05) |

Continued

| Model | DGP 1: $\beta_3 = 0$ | | DGP 2: $\beta_3 = -3$ | | DGP 3: $\beta_3 = -3$ | | DGP 4: $\beta_3 = -3$ | |
|---|---|---|---|---|---|---|---|---|
| OLS SACS | 0.074 | (2E-05) | 0.074 | (2E-05) | 0.077 | (2E-05) | 0.078 | (2E-05) |
| OLS ANCOVA | 0.07 | (2E-05) | 0.065 | (2E-05) | 0.072 | (2E-05) | 0.059 | (2E-05) |
| OLS post | 0.07 | (2E-05) | 0.07 | (2E-05) | 0.072 | (2E-05) | 0.055 | (2E-05) |
| Tobit SACS | 0.13 | (5E-05) | 0.13 | (5E-05) | 0.14 | (6E-05) | 0.15 | (6E-05) |
| Tobit ANCOVA | 0.14 | (6E-05) | 0.14 | (6E-05) | 0.15 | (6E-05) | 0.13 | (6E-05) |
| Tobit post | 0.15 | (7E-05) | 0.15 | (6E-05) | 0.16 | (7E-05) | 0.13 | (6E-05) |

**Table 1** Continued

ANCOVA, analysis of covariance; DGP, data generating process; MLM, multilevel model; OLS, ordinarily least squares; SACS, Simple Analysis of Change Scores.

is poor, suggesting bias in model based SE, that is, they are too small.

### NJR of England, Wales, Northern Ireland and the Isle of Man

Following application of inclusion and exclusion criteria, there were 162 513 patients with pre-operative and post-operative OHS available for analysis. Figure 6 illustrates the results of the exemplar dataset using different approaches while attempting to estimate the effect of BMI category on the efficacy of surgery, whereas figures 7 and 8 illustrate the use of RCS to assess the same question.

### Exemplar analysis

A single-level OLS SACS appoach suggests a positive association between BMI and change in OHS, that is, patients with greater BMIs have greater gains in well-being, whereas OLS ANCOVA and OLS post-score models suggest a negative association. The single-level post-model score is approximately 50% greater than the ANCOVA model. All single-level Tobit models suggest a negative association between BMI and OHS. The Tobit SACS model is the smallest, with both the Tobit ANCOVA and post-models estimating substantially larger effects. The constrained ML Tobit models all provide equivalent (to two decimal places) results, suggesting there

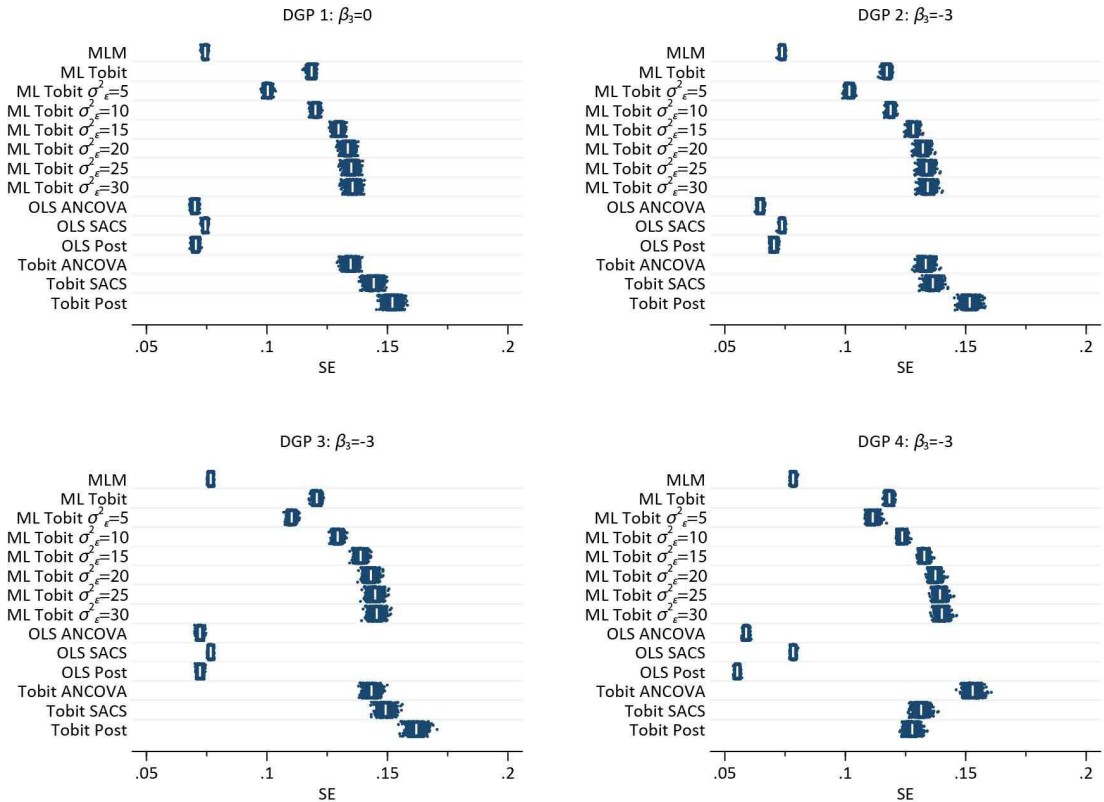

**Figure 5** Plot of 1000 estimated SEs by each DGP, for each method of analysis. Within each method, the vertical axis is the repition number of each simulated dataset. The white pipe symbol is the average of the SEs. ANCOVA, analysis of covariance; DGP, data generating process; MLM, multilevel model; OLS, ordinarily least squares; SACS, Simple Analysis of Change Scores.

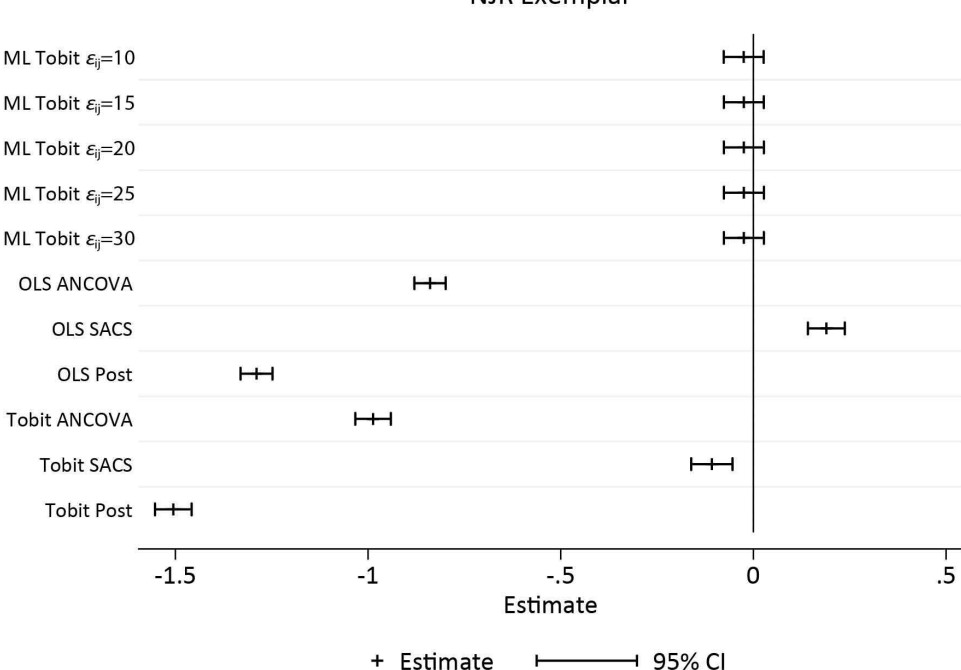

**Figure 6** Estimate and 95% CIs of constrained ml Tobit, Single-level OLS and Tobit: ANCOVA, sacs and post-models. ANCOVA, analysis of covariance; ML, multilevel; NJR, National Joint Registry; OLS, ordinarily least squares; SACS, Simple Analysis of Change Scores.

is no effect of BMI on the change in OHS pre- and post-surgery, see figure 6.

### RCS approach

Crude analyses, which model BMI using RCS, illustrate a complex association between BMI and pre-operative OHS. A~4.5 point reduction in OHS is observed as BMI increases between 20 and $50 \, kg/m^2$. However, the change in OHS between pre-surgery and post-surgery is very weakly associated with pre-operative BMI, with individuals with BMIs $<25 \, kg/m^2$ and $>45 \, kg/m^2$ receiving modestly greater gains than those patients with an average BMI of $28 \, kg/m^2$. However, with less than ½ a unit variation across the range of BMI observed in the cohort, the difference falls well below anything that could be considered clinically meaningful, see figure 7. Following adjustment for patient factors, operation factors, centre factors, consultant factors and deprivation, there was little difference in the pattern of change compared with crude results, see figure 7. Single-level approaches are illustrated in figure 8, with OLS and Tobit models giving similar patterns of results. ANCOVA and the post-model specification suggest a strong inverse association with BMI, with obese individuals receiving less improvement following surgery. OLS SACS indicates that obesity is associated with greater gains in OHS following surgery. Conversely, Tobit SACS models indicate that obesity is associated with smaller gains in OHS following surgery.

### DISCUSSION

The results of the simulation study clearly illustrate that, in the presence of floor and ceiling effects, neither baseline adjustment, or SACS will yield unbiased estimates of the effect of an exposure on the outcome of interest. Single-level modifications to account for floor and ceiling effects such as the Tobit model only work in the context of a randomised trial, that is, when there is no difference between baseline values by BMI. Importantly, single-level methods, OLS and Tobit models, induce significant bias, with negligible coverage, when $\beta_3 = 0$, that is, there is no change in the pre- post-surgery well being by BMI. Fully identified MLTM with three measurement occasions, return unbiased estimates with coverage close to advertised levels. In pre- post-designs with two measurement occasions ML Tobit models, with constrained level 1 variances, return estimates very close to being unbiased, but coverage is less than advertised indicating bias in the model based SEs.

The simulation study is consistent with a lay intuition with respect to analyses of floor and ceiling effects. Assuming we accept that either the MLM and OLS change analyses are appropriate in the absence of floor and ceiling effects, DGP 1 illustrates that when there is no effect of obesity on the efficacy of surgery, the addition of an artificial ceiling compresses the gain of individuals towards the top of the distribution. Due to the baseline association between obesity and well-being, underweight individuals tend to have gains that are more compressed compared with obese individuals. This inevitably induces bias, and provides evidence of a change in presurgery and post-surgery well-being by BMI, where none actually exist. Similarly, in DGP 2 (no baseline differences) where there is truly an interaction effect, will also lead to biased estimates. The DGP used in the simulation assumes

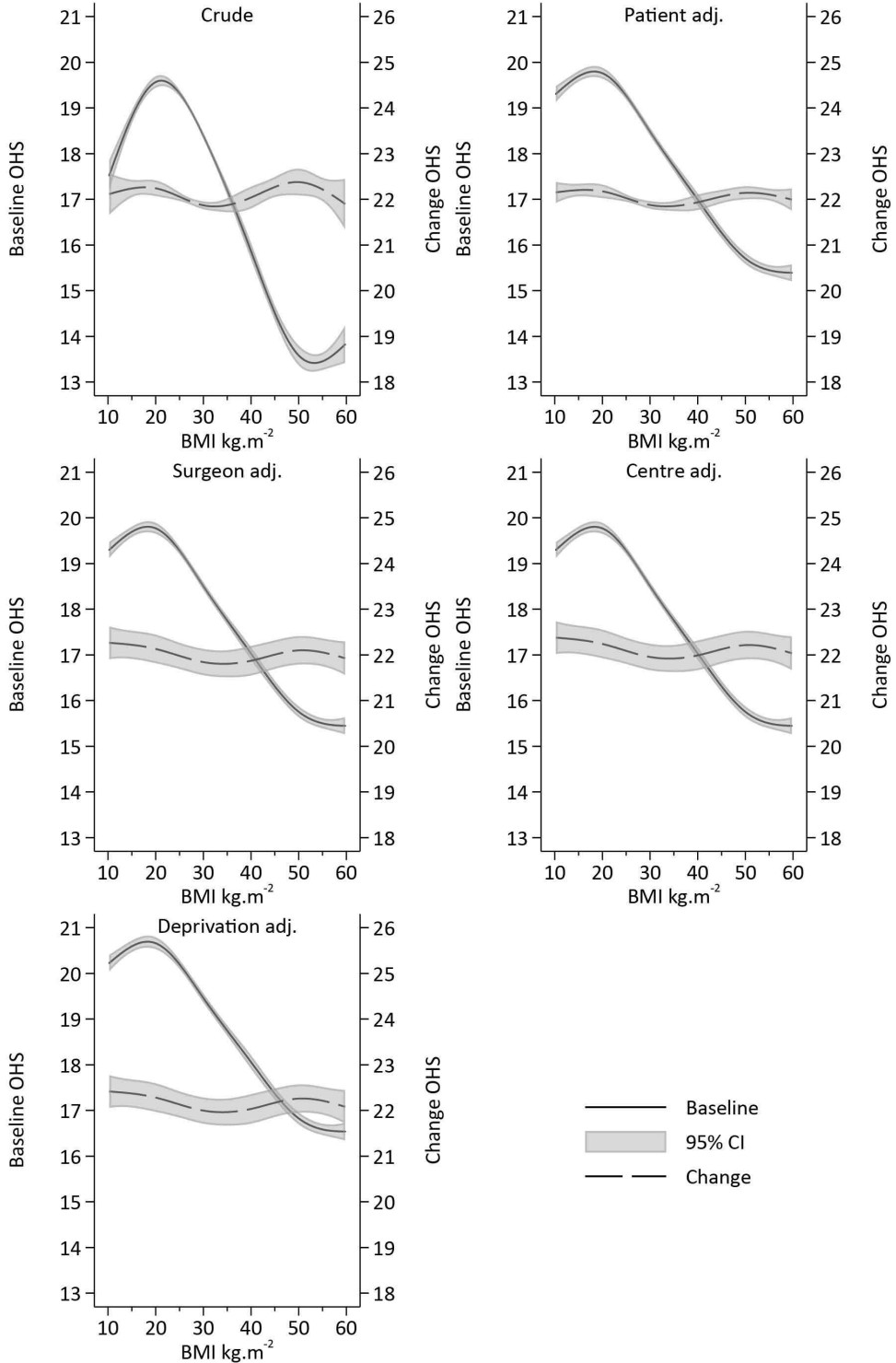

**Figure 7** Estimates and 95% CIs of baseline and change in Oxford Hip Score (OHS) pre- post-surgery and its association with body mass index (BMI) adjusted for confounding.

underweight individuals benefit more from surgery than heavier individuals, which results in a fanning out of the trajectories. Underweight individuals have truly greater gains than obese individuals, but these gains are underestimated due to the ceiling effect, resulting in bias towards the null. In DGPs 3 and 4 (baseline differences in BMI, and interaction between BMI and change), we see a more extreme pattern of results compared with DGP

2, but overall consistency with the expected response of compressing individual gains which have initially higher starting values.

In the exemplar analysis of NJR data, the pattern of results is very similar to that of DGP 1 of the simulation, suggesting that results of the simulation are likely to be replicated in real-world datasets. The more comprehensive analysis of the NJR data, using RCS to reflect the continuous nature of BMI,

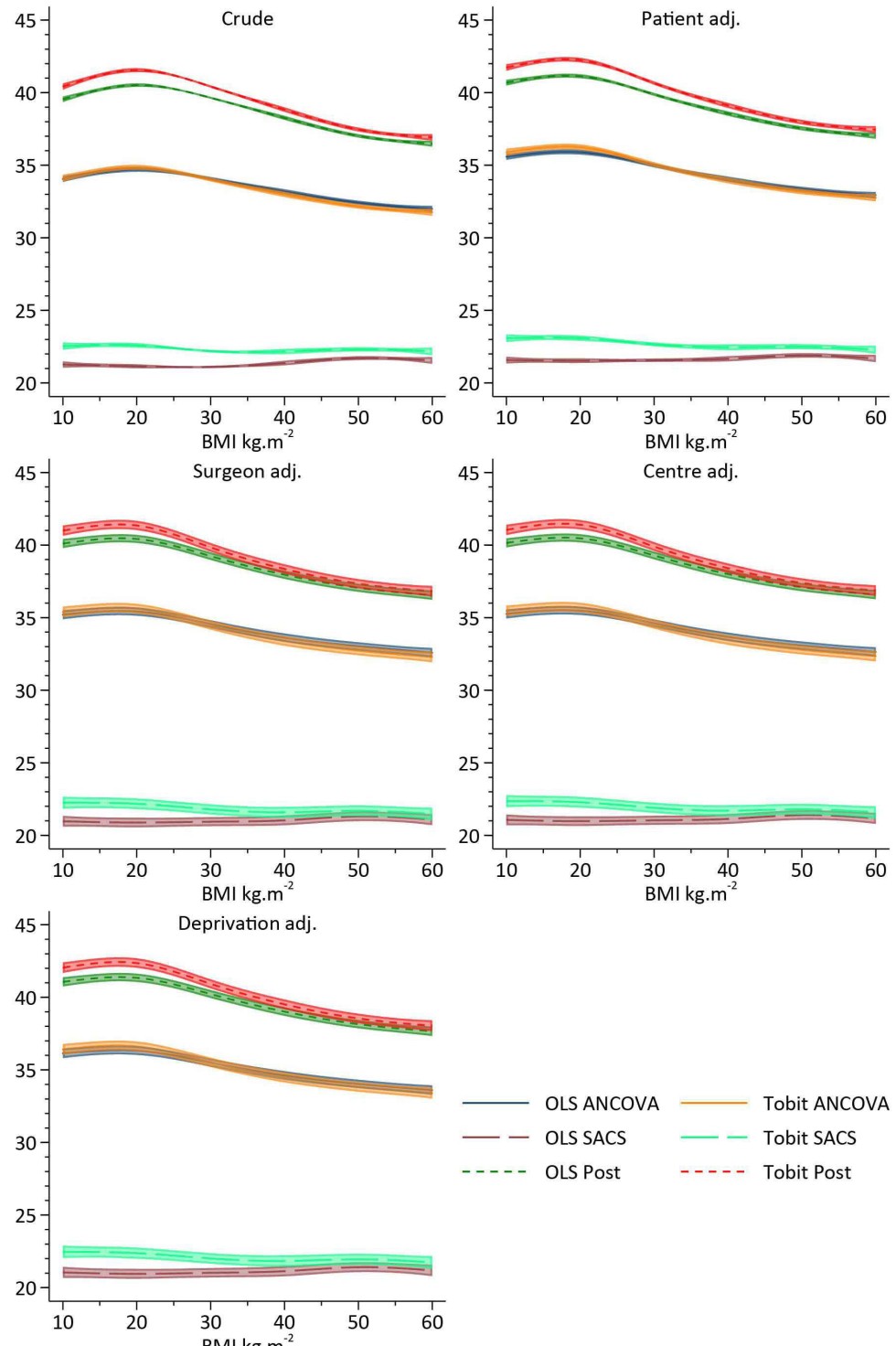

**Figure 8** Estimates and 95% CIs of single-level approaches to the analysis of change in Oxford Hip Score pre- post-surgery and its association with body mass index (BMI) adjusted for confounding. ANCOVA, analysis of covariance; OLS, ordinarily least squares; SACS, Simple Analysis of Change Scores.

aptly illustrate where the effects from misspecified single-level models are arising from. The ML Tobit model illustrates a strong negative association between BMI and pre-operative OHS, and failing to account for these baseline differences appropriately when attempting to estimate change leads to variation at baseline being incorporated in the estimate of change. Furthermore, the ability to adjust both baseline and

post-surgical OHS for their pronounced floor and ceiling effects, respectively, leads to unbiased estimates of the effect of interest. Unfortunately, due to the constraints on the level 1 variance, interpretation of the random effects are difficult, as they depend on the magnitude of the variance applied in the constraint, see online supplementary figure 16. However, the models clearly illustrate that change in PROMS following

THR do not depend on BMI, and surgery appears to be effective for patients regardless of their BMI.

## CONCLUSION

Floors and ceilings in PROM instruments have somewhat predictable effects on estimated coefficients from standard OLS models that do not adjust for floor or ceiling effects, assuming the true underlying association is known. As this is rarely the case, it is important to consider a variety of different DGP to explore the likely impact on an analysis. It is important to consider the validity of the assumptions underpinning the Tobit model, that is, that the latent response is truly continuous and that there is a true ceiling just beyond the range of the measurement being used.

Single-level Tobit models do not ameliorate floor and ceiling effects in SACS. However, ML Tobit models appear to recover the effects of interest under specific assumptions. The analyses of pre- post-designs require further constraints to ensure models are fully identified. The difference between analytical approaches can profoundly alter the interpretation of the model parameters, and this may have serious consequences if used to generate policy inappropriately. For example, inappropriate analyses that fail to consider DGP appropriately may lead to the restriction of joint replacement for overweight or obese patients.

When designing a study to investigate the effect of an exposure on change in health status, it would be preferable to use a measurement instrument that does not have floor or ceiling effects as inference is less complicated, and design trumps analysis in most scenarios. If the use of measurement instrument with floor and ceiling effects is unavoidable, it is preferable to collect data at three time points which ensure models are fully identified, alleviating the need to constrain level 1 variance in order to identify models, again design trumps analysis. If retrospective analysis of pre- and post-data sets are required, it appears that using ML Tobit model with constrained level 1 error variance would be preferable to single-level approaches.

Broadly speaking the analyses of this simulation are in agreement with the work of Glymour *et al*, that analysis of change and its interaction with an exposure at baseline, should not be adjusted for baseline measurements in observational data. The presence of floor and ceiling effects in data requires additional assumptions which makes things marginally more complex.

**Author affiliations**
[1]Musculoskeletal Research Unit, Bristol Medical School, 1st Floor Learning & Research Building, Southmead Hospital, University of Bristol, Bristol, BS10 5NB, UK
[2]Population Health Sciences, Bristol Medical School, Canynge Hall, 39 Whatley Road, University of Bristol, Bristol, BS8 2PS, UK
[3]National Institute for Health Research Bristol Biomedical Research Centre, University of Bristol, Bristol, UK
[4]Norwich Medical School, University of East Anglia, Norwich, UK

**Acknowledgements** We thank the patients and staff of all the hospitals who have contributed data to the National JointRegistry. We are grateful to the Healthcare Quality Improvement Partnership, the National JointRegistry Steering Committee, and staff at the National Joint Registry for facilitating this work. Theviews expressed represent those of the authors and do not necessarily reflect those of the NationalJoint Registry Steering Committee or Healthcare Quality Improvement Partnership, who do not vouchfor how the information is presented.

**Contributors** AS, MRW, AJ, AJM, AWB and YB-S were responsible for the study design, AS conducted the data analysis. AS, MRW, AJ, AJM, AWB and YB-S were responsible for interpreting the data. AS, MRW, AJ, AJM, AWB and YB-S prepared and edited and approved the final manuscript.

**Funding** AS is funded by an MRC Strategic Skills Fellowship MR/L01226X/1. This study was supported by the NIHR Biomedical Research Centre at University Hospitals Bristol NHS Foundation Trust and the University of Bristol.

**Disclaimer** The views expressed represent those of the authors and do not necessarily reflect those of the National Joint Registry Steering Committee or Healthcare Quality Improvement Partnership, who do not vouch for how the information is presented.

**Patient and public involvement** Patients and/or the public were involved in the design, or conduct, or reporting, or dissemination plans of this research. Refer to the Methods section for further details.

**Patient consent for publication** Not required.

**Ethics approval** Ethics approval of pseudo anonymised analysis of NJR data is considered as secondary use of clinical registry data, under HRA guidance this does not require formal ethical approval. However, all research projects are internally approved by the NJR. The full NJR privacy notice can be found online (http://www.njrcentre.org.uk/njrcentre/About-the-NJR/Privacy-Notice-GDPR).

**Provenance and peer review** Not commissioned; externally peer reviewed.

**Data availability statement** Data may be obtained from a third party and are not publicly available. Access to the data can be made via research requests to the National Joint Registry of England, Wales, Northern Ireland and the Isle of Man. Full details can be found at http://www.njrcentre.org.uk/njrcentre/Research/Research-requests.

**ORCID iDs**
Adrian Sayers http://orcid.org/0000-0001-7452-5043
Michael R Whitehouse http://orcid.org/0000-0003-2436-9024

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
