## [Reviewer comments · BMJ Open]

ARTICLE DETAILS

TITLE (PROVISIONAL)	Analysis of change in patient reported outcome measures with floor and ceiling effects using the multi-level Tobit model: A simulation study and an example from a National Joint Register using body mass index and the Oxford Hip Score.
AUTHORS	Sayers, Adrian; Whitehouse, Michael; Judge, Andrew; MacGregor, Alex; Blom, AW; Ben-Shlomo, Yoav

VERSION 1 – REVIEW

REVIEWER	R Zwiars University of Amsterdam, the Netherlands
REVIEW RETURNED	16-Nov-2019

GENERAL COMMENTS	I have read the manuscript with great interest. I think the authors address an important topic, since PROMs are used increasingly in outcome studies and analysis are often inadequate. The paper is well written, the research question is relevant and the conclusion is clear and supported by the results. As far as my knowledge on this type of studies goes, I think the methodology of this paper is sound. However, the statistical analyses used in this study are far beyond my knowledge and therefore should be reviewed by a statistician.
---

REVIEWER	Eric Bohm University of Manitoba Canada
REVIEW RETURNED	19-Nov-2019

GENERAL COMMENTS	None.
-------

REVIEWER	Dr. Sajjad Ahmad Khan Department of Statistics, Islamia College Peshawar, Pakistan.
REVIEW RETURNED	12-Feb-2020

GENERAL COMMENTS	General Comments: The novelty of the research paper is very good. Paper consists of different sections such as Introduction, Methods, Models, Data analysis, Results and Discussions. The paper is written on 46 pages altogether, and enriched by number of tables and figures. The bibliographic part was validated with 23 references. Summary:
---

	The author(s) used a linear multi-level model and a multi-level Tobit model (MLTM) in the following three scenarios  1. A simple model for post-surgery well-being 2. A Simple Analysis of Change Score (SACS) and 3. A model for change adjusted for baseline i.e. baseline adjusted ANCOVA. The performance of estimates in all approaches was measured through bias, coverage, empirical standard error, model based standard error, mean square error, relative error and relative precision. Findings Their findings suggest that in simple analysis of change scores, Single-level Tobit models do not improve floor and ceiling effects. Conversely, ML Tobit models appear to recover the effects of interest under specific assumptions. Similarly, ML Tobit model with constrained level 1 error variance would be preferable to single-level approaches in case of retrospective analysis of pre-post data sets. Moreover, their analyses seem to be genuine as the simulation results are supported by real data analysis also. Recommendations The paper is suitable for publication
--	--

VERSION 1 – AUTHOR RESPONSE

We would like to thank the editor and reviewers for taking their time to review this manuscript.

In accordance to the editorial requests we have:

- 1) Revised the strengths and limitations of the manuscript.
- 2) Included a statement on the ethical approval for the basis of the study.
- 3) Revised the objectives of abstract.
- 4) Alexander J Macgregor is the appropriate presentation of the authors' name.